# Spatiotemporal Patterns of Non-Communicable Disease Mortality in the Metropolitan Area of the Valley of Mexico, 2000–2019

**DOI:** 10.3390/diseases13080241

**Published:** 2025-08-01

**Authors:** Constantino González-Salazar, Kathia Gasca-Gómez, Omar Cordero-Saldierna

**Affiliations:** 1ICAyCC—Instituto de Ciencias de la Atmósfera y Cambio Climático, UNAM—Universidad Nacional Autónoma de México, Mexico City 04510, Mexico; kathia.gasca.gomez@gmail.com (K.G.-G.); omar15cordero200105@gmail.com (O.C.-S.); 2C3—Centro de Ciencias de la Complejidad, Universidad Nacional Autónoma de México, Mexico City 04510, Mexico; 3LANBIOCC—Laboratorio Nacional de Biología del Cambio Climático, SECIHTI, Mexico City 04510, Mexico

**Keywords:** chronic diseases, spatial epidemiology, health determinants, mortality hotspots, non-communicable diseases

## Abstract

Background: Non-communicable diseases (NCDs) are a leading cause of mortality globally, contributing significantly to the burden on healthcare systems. Understanding the spatiotemporal patterns of NCD mortality is crucial for identifying vulnerable populations and regions at high risk. Objectives: Here, we evaluated the spatiotemporal patterns of NCD mortality in the Metropolitan Area of the Valley of Mexico (MAVM) from 2000 to 2019 for five International Classification of Diseases chapters (4, 5, 6, 9, and 10) at two spatial scales: the municipal level and metropolitan region. Methods: Mortality rates were calculated for the total population and stratified by sex and age groups at both spatial scales. In addition, the relative risk (RR) of mortality was estimated to identify vulnerable population groups and regions with a high risk of mortality, using women and the 25–34 age group as reference categories for population-level analysis, and the overall MAVM mortality rate as the reference for municipal-level analysis. Results: Mortality trends showed that circulatory-system diseases (Chapter 9) are emerging as a concerning health issue, with 45 municipalities showing increasing mortality trends, especially among older adults. Respiratory-system diseases (Chapter 10), mental and behavioral disorders (Chapter 5) and nervous-system diseases (Chapter 6) predominantly did not exhibit a consistent general mortality trend. However, upon disaggregating by sex and age groups, specific negative or positive trends emerged at the municipal level for some of these chapters or subgroups. Endocrine, nutritional, and metabolic diseases (Chapter 4) showed a complex pattern, with some age groups presenting increasing mortality trends, and 52 municipalities showing increasing trends overall. The RR showed men and older age groups (≥35 years) exhibiting higher mortality risks. The temporal trend of RR allowed us to identify spatial mortality hotspots mainly in chapters related to circulatory, endocrine, and respiratory diseases, forming four geographical clusters in Mexico City that show persistent high risk of mortality. Conclusions: The spatiotemporal analysis highlights municipalities and vulnerable populations with a consistently elevated mortality risk. These findings emphasize the need for monitoring NCD mortality patterns at both the municipal and metropolitan levels to address disparities and guide the implementation of health policies aimed at reducing mortality risk in vulnerable populations.

## 1. Introduction

Non-communicable diseases (NCDs) are a leading global health concern, accounting for approximately 70% of deaths worldwide [1,2]. The growing burden of NCDs is a major priority of the global health agenda, as they cause more deaths than all other causes combined [3]. NCDs are conditions with no known causative agents and are generally not transmissible from one person to another. They are usually chronic in nature, with slow onset and lengthy progression, generating costs that go beyond health, perpetuating poverty and undermining quality of life [1,4,5]. The highest burden of NCDs can be attributed to chronic diseases such as cardiovascular diseases, cancers, chronic respiratory illnesses, and diabetes [1,6].

Death rates from NCDs are higher in low- and middle-income countries, and, at least in high-income countries, among people with lower socioeconomic status [7,8,9], making NCDs a significant barrier to reducing both global and national health inequalities [9,10]. For decades, NCDs were mainly a concern of high-income countries; nowadays, they represent the primary problem in the developing world. These trends reflect growing populations, rapid unplanned urbanization, and unhealthy behaviors [11]. Furthermore, NCDs affect productive age groups, and adults in low- and lower-middle-income countries have double the risk of dying from an NCD compared with adults in high-income countries, creating a vicious cycle of poverty and ill-health [3].

As a result of prioritizing NCDs, target 3.4 of the Sustainable Development Goals (SDGs) was introduced to reduce the total NCD mortality rate by one third by 2030 [2]. This global commitment reflects the growing recognition of NCDs as a major public health challenge, especially in low- and middle-income countries, where health systems are often ill-equipped to address their long-term burden [9]. Achieving this goal requires not only clinical interventions but also a comprehensive understanding of spatial and demographic disparities in NCD-related mortality.

NCD-related deaths impose a significant burden upon communities and healthcare systems. The challenge arises when the rate of health transition outpaces the development of health services [12]. Therefore, health planners and policymakers need to have a deep understanding of the epidemiological dynamics of NCDs in order to implement optimal resource allocations and provide appropriate health services. Consistent with global trends, Mexico has experienced a rise in the prevalence of NCDs in recent years; from 2006 to 2016, the national prevalence of diabetes mellitus increased from 7.2 to 9.4%. Overweight and obesity prevalence increased from 69.7% in 2006 to 72.5% in 2016 [13,14]. Health care costs associated with obesity-related NCDs are projected to reach USD 1.2 billion by 2030 and USD 1.7 billion by 2050 in Mexico [15]. Overall, healthcare service providers in Mexico are spending a significant proportion of their health expenditure on the financial burden of NCDs [14]. Although measuring the spatial and temporal trends of NCD-related mortality is very important to improve the existing healthcare systems, a comprehensive estimate considering social and environmental determinants is lacking in Mexico.

In this study, we aimed to assess the spatiotemporal patterns of mortality from non-communicable diseases in the Metropolitan Area of the Valley of Mexico (MAVM), one of the largest urban population centers in the world. This demographic concentration is the outcome of historical processes that have shaped the region into the country’s primary industrial and economic hub [16]. Despite its economic prominence—with the highest national Gross Domestic Product (GDP) and 36 industrial parks as of 2023—social conditions across the MAVM remain highly unequal [17]. According to the 2020 national census, 6% of the population lived in extreme poverty and 37% in moderate poverty. The most prevalent forms of social deprivation were lack of access to social security and healthcare services [18].

Geographically, the MAVM lies in a semi-enclosed basin, with an average valley floor elevation of 2240 m above sea level and surrounding mountains rising to approximately 3200 m [19]. Surrounding mountains—especially those to the south—form a natural barrier that limits the influence of coastal climates. Thermal inversions occur annually on approximately 70% of the days, and the topography further hinders the dispersion of atmospheric pollutants [19].

These environmental, demographic, and social characteristics are critical when analyzing NCD mortality patterns through the lens of social determinants of health. Moreover, the region’s rapid demographic growth and economic activities have placed increasing pressure on ecological conservation zones and have impacted the overall quality of life for its residents [20].

Our analysis was conducted at two spatial scales: the municipal level and the entire metropolitan region. We integrated key health determinants, such as age and sex, to identify vulnerable population groups and high-risk areas. By examining both local and regional trends over time, this study provides a comprehensive overview of the NCD burden. This approach could support the design of evidence-based public health policies and the implementation of targeted interventions across diverse demographic and geographic contexts within the MAVM.

## 2. Materials and Methods

### 2.1. Study Area

The study area comprised the Metropolitan Area of the Valley of Mexico, which included 16 boroughs of Mexico City, 59 municipalities of the State of Mexico, and one municipality of Hidalgo, and is located at 2240 m above sea level [20] (Figure 1). Covering an area of approximately 7845 km^2^, the MAVM is home to 21.8 million people [18], making it one of the most densely populated urban regions in the world. It is also the country’s most important economic hub, generating the highest Gross Domestic Product (GDP) nationally [16]. The region’s demographic and socioeconomic diversity, coupled with its unique geographical features—such as its high elevation and semi-enclosed basin topography—makes it a critical setting for investigating health inequities and the spatial distribution of non-communicable disease mortality.

### 2.2. Mortality Data

Mortality data were obtained from the open-data portal of the Ministry of Health of Mexico (http://www.dgis.salud.gob.mx/contenidos/basesdedatos/da_defunciones_gobmx.html, accessed on 17 November 2024) for the period 2000–2019. We selected records where cause of death was classified within five chapters of the International Classification of Diseases, 10th Revision (ICD-10) [21]: Chapter 4 (Endocrine, Nutritional, and Metabolic Diseases), Chapter 5 (Mental, Behavioral, and Neurodevelopmental disorders), Chapter 6 (Diseases of the Nervous System), Chapter 9 (Diseases of the Circulatory System), and Chapter 10 (Diseases of the Respiratory System). Data were filtered to include only deaths where the place of occurrence coincided with the individual’s place of residence, ensuring all cases belonged to residents within our study area.

### 2.3. Population Data

To facilitate a meaningful comparison of mortality for population groups and regions, we calculated mortality rates standardized per 100,000 inhabitants. These rates were estimated using sex-, age-, and municipality-specific demographic data. Annual population data spanning the period from 2000 to 2019 were obtained from the Consejo Nacional de Población (CONAPO) of Mexico [22].

### 2.4. Data Analysis

Deaths were categorized into eight age groups with 10-year intervals, except the first and the last: 0–4, 5–14, 15–24, 25–34, 35–44, 45–54, 55–64, and 65+ years. Mortality rates (MR) were calculated by dividing the number of deaths in each age group or sex by the total population, and then standardizing per 100,000 inhabitants:(1)$$MR=\frac{D_{i}}{P_{i}}\times 100,000$$
where D_i_ is the number of deaths of each age group or sex, and P_i_ is the total population by age or sex. Standardization was performed using population data from CONAPO, which included both total population and sex-specific data. This approach enabled consistent comparisons across municipalities, years, and population subgroups, providing a comprehensive analysis of mortality trends by sex, age group, and region.

#### 2.4.1. Temporal Trends

To assess temporal trends in non-communicable disease mortality at the municipal level, we applied linear regression models to mortality rates over the period 2000–2019. For each municipality and disease type, a linear model was fitted with the year as the predictor and mortality rate as the response variable. The slope coefficient (*β*) was used to determine the direction and magnitude of the trend over time. Trends were classified into three categories: positive, negative, and no trend, based on the sign and statistical significance of the slope. A positive slope (*β* > 0.5 and *p* < 0.05) was classified as a positive trend, while a negative slope (*β* < −0.5 and *p* < 0.05) indicated a negative trend. Otherwise, the trend was labeled as no trend. The threshold for the slope coefficient (0.5 or −0.5) was defined based on the scale of the rates (i.e., per 100,000 inhabitants) to ensure sensitivity to subtle but meaningful changes. This approach ensures that only statistically significant changes in mortality rates are classified as trends. The analysis was conducted at both the municipal level and the MAVM level by aggregating mortality and population data. For both spatial levels, mortality trends were analyzed for the total population, stratified by sex (men and women) and by age groups.

##### Autocorrelation Assessment

Given the time-series nature of our mortality data, temporal autocorrelation in the residuals of all linear regression models was rigorously evaluated. This assessment was conducted for every fitted model, including those representing trends at the Metropolitan Area of the Valley of Mexico level (for the total population and stratified by sex and age group), as well as for each individual municipality (also stratified by total population, sex, and age group).

Due to the large number of models generated in this comprehensive analysis, the primary method used to assess autocorrelation was the formal Ljung–Box test. For each model, this test was applied to the residuals to statistically detect significant autocorrelation, considering up to 10 lags. Models with an Ljung–Box test *p*-value of less than 0.05 were classified as exhibiting statistically significant autocorrelation. Additionally, autocorrelation function (ACF) and partial autocorrelation function (PACF) plots (correlograms) were visually examined for aggregated models (i.e., at the MAVM level by disease category) to complement the quantitative results and highlight common patterns.

#### 2.4.2. Relative Risk Estimation by Age and Sex Groups

To assess demographic differences in mortality, we estimated relative risks (RR) by sex and age groups, using women and the 25–34 age group as reference categories. This approach allowed the comparison of mortality patterns across demographic groups while accounting for population size differences [23,24,25].

Sex-specific relative risks were estimated annually as the ratio of men to women mortality rates (MR):(2)$${RR}_{sex}=\frac{{MR}_{men}}{{MR}_{women}}$$

For age groups, relative risks for each year were calculated as the ratio of the mortality rate in a given age group to that in the 25–34 years reference group:(3)$${RR}_{age}=\frac{{MR}_{age}}{{MR}_{25–34}}$$

Standard errors and 95% confidence intervals (*CI*) were estimated on the log scale assuming Poisson-distributed death counts [23]:(4)$${SE}_{log(RR)}=\sqrt{\frac{1}{D_{target}}+\frac{1}{D_{reference}}}$$(5)$$CI=exp(log(RR)\pm 1.96\times {SE}_{log(RR)})$$
where D is the number of deaths in the target and reference groups. All RR values were visualized using line plots, with shaded ribbons indicating 95% confidence intervals. For age-group analysis, we applied a log scale to the *y*-axis to enhance interpretability across widely varying RR magnitudes. For sex comparisons, both male and female RR values were presented side-by-side, facilitating a more nuanced understanding of sex-specific mortality trends

#### 2.4.3. Relative Risk Estimation by Municipalities

To assess spatial variations in mortality risk, we estimated the municipality-specific relative risk (RR) of mortality by using the mortality rate of the MAVM as the reference rate.(6)$${RR}_{mun}=\frac{{MR}_{mun}}{{MR}_{MAVM}}$$

Standard errors and a 95% confidence interval (*CI*) were estimated on the log scale assuming Poisson-distributed death counts:(7)$${SE}_{log(RR)}=\sqrt{\frac{1}{D_{mun}}+\frac{1}{D_{MAVM}}}$$(8)$$CI=exp(log(RR)\pm 1.96\times {SE}_{log(RR)})$$
where D_mun_ is the number of deaths in each municipality, and D_MAVM_ is the total deaths registered in the MAVM.

RR and their corresponding 95% CI were calculated for each disease chapter from 2000 to 2019. RR values were spatially mapped by assigning to each municipality polygon its corresponding RR estimate. To highlight areas with elevated mortality risk, we visualized only municipalities with statistically significant high RR values, defined as those with an RR > 1 and a lower bound of the 95% confidence interval (LCI) > 1. This strict criterion ensured that only municipalities where there is at least 95% confidence that the mortality risk is truly elevated were identified as hotspots. Additionally, we quantified the persistence of excess mortality risk by counting the number of years each municipality exhibited a statistically significant RR for each disease chapter. The resulting frequency maps reveal municipalities with consistently elevated mortality risks throughout the study period.

#### 2.4.4. Statistical Software

All statistical analyses were performed using R software (version 4.3.3). Data manipulation and processing were conducted with the *dplyr* package. Linear regression models were fitted using base R functions. To assess autocorrelation in the residuals, functions from the *lmtest* package—specifically Box.test()—were employed. Visualization of autocorrelation functions (ACF) and partial autocorrelation functions (PACF) was performed using base R functions (acf() and pacf()). Model outputs were extracted and organized with the *broom* package, and iteration over multiple analysis groups was handled using *purrr*. Data visualizations were created with the *ggplot2* package.

## 3. Results

Mortality data spanning the period from 2000 to 2019 were obtained for the five NCD chapters (Chapters 4, 5, 6, 9, and 10). A total of 747,131 deaths were recorded during the study period. Chapter 9 (Diseases of the Circulatory System) accounted for the highest number of deaths (348,201), while Chapter 5 (Mental, Behavioral, and Neurodevelopmental Disorders) had the lowest number. Overall, the majority of deaths occurred among individuals aged 35 years and older, and deaths were generally higher among men compared to women. However, Chapter 9 showed a distinct sex-specific pattern, with women exhibiting a greater number of deaths. Additionally, for Chapter 10 (Diseases of the Respiratory System), the 0–4 age group registered a notably high number of deaths, surpassed only by the 65+ age group, underscoring the particular vulnerability of young children to respiratory conditions (Table 1). Mortality data are provided in Appendix A. Spatiotemporal mortality patterns at both the metropolitan and municipal levels for each NCD chapter are presented separately in the following sections.

### 3.1. Mortality by Non-Communicable Diseases at the Metropolitan Scale in the MAVM

#### 3.1.1. Mortality Trends for the Total Population

Figure 2 shows the temporal trends in mortality rates (per 100,000 inhabitants) from 2000 to 2019 across the five analyzed disease chapters. A statistically significant upward trend was observed in Chapter 4 and Chapter 9, with estimated slopes of *β* = 0.24 (*R*^2^ = 0.96, *p* = 7.5 × 10^−14^) and *β* = 0.40 (*R*^2^ = 0.96, *p* = 1.2 × 10^−13^), respectively. These findings indicate a consistent and pronounced increase in mortality over the study period for both chapters. For Chapter 4, the Ljung–Box test showed no significant autocorrelation in the residuals (*Q** = 3.94, *df* = 4, *p* = 0.4142), supporting the adequacy of the linear model. The ACF revealed a modest spike at lag 1, suggesting some short-term correlation, but all subsequent lags remained within the confidence bounds. The PACF showed a moderate value at lag 1 and a minor dip at lag 8, both within acceptable limits (Appendix A). Together, these results suggest that the temporal structure was adequately captured by the model.

In contrast, for Chapter 9, the Ljung–Box test indicated statistically significant autocorrelation in the residuals (*Q** = 16.139, *df* = 4, *p* = 0.0028), indicating that the linear model does not fully capture the temporal structure. The ACF exhibited strong positive autocorrelation at lags 1 and 2, along with significant negative autocorrelations at lags 5, 7, and 8, indicating potential autoregressive behavior. The PACF also showed a pronounced spike at lag 1 and a smaller peak at lag 2, while most subsequent lags stayed within the confidence limits (Appendix A). These patterns highlight the presence of residual temporal dependence and suggest the need for a more complex model specification. Nonetheless, given the strong and highly significant linear trend and considering the exploratory nature of the analysis, the linear model was retained to highlight the overall increase in mortality and ensure comparability across chapters.

Chapters 6 and 10 exhibited weaker yet statistically significant increases in mortality rates, with slopes of *β* = 0.01 (*R*^2^ = 0.57, *p* = 0.00013) and *β* = 0.03 (*R*^2^ = 0.45, *p* = 0.0013), respectively, suggesting a slower but steady upward trend. The Ljung–Box test for Chapter 6 revealed statistically significant autocorrelation (*Q** = 9.8001, *df* = 4, *p* = 0.0439), which was consistent with the correlogram results the ACF and PACF showed, with mild but significant spikes at lag 1 (Appendix A). For Chapter 10, however, the Ljung–Box test indicated no significant autocorrelation (*Q** = 2.432, *df* = 4, *p* = 0.657), and both the ACF and PACF plots displayed no prominent spikes (Appendix A), indicating the linear model was appropriate for describing its trend. Chapter 5 showed no statistically significant change in mortality over the study period (*β* = 0.00, *R*^2^ = 0.15, *p* = 0.087), which suggests relative stability in mortality rates during the period analyzed. The Ljung–Box test did not show significant autocorrelation (*Q** = 8.1955, *df* = 4, *p* = 0.0847). However, correlograms indicated more nuanced temporal structures. In the ACF, the bar at lag 1 was high and exceeded the upper confidence bound, indicating a strong positive autocorrelation with the previous year. Lag 2 surpassed the confidence threshold but remained close to the bounds. In the PACF, the bar at lag 1 also surpassed the upper confidence limit, suggesting a direct temporal dependency, while all subsequent lags remained within the confidence intervals (Appendix A). These patterns warrant further exploration beyond linear modeling approaches.

#### 3.1.2. Sex-Specific Mortality Trends

We analyzed temporal mortality trends by disease chapter, stratified by sex. Significant increasing trends (positive *β*) were observed in both sexes for Chapter 4 and Chapter 9, suggesting a growing burden of these NCDs in the region. For Chapter 9, significant autocorrelation was detected in the residuals for both men (*Q** = 30.099, *df* = 10, *p* = 0.0008) and women (*Q** = 42.122, *df* = 10, *p* < 0.00001), indicating that while linear models capture the overall upward trends, some temporal dependence remains in the unexplained variance. In contrast, for Chapter 4, residuals for men (*Q** = 15.319, *df* = 10, *p* = 0.121) and women (*Q** = 12.141, *df* = 10, *p* = 0.276) did not exhibit significant autocorrelation, indicating that the linear models adequately capture the temporal dynamics (Appendix A).

No clear trends were identified for Chapters 5, 6, and 10 in either sex. However, for Chapter 6, a slight upward shift was detected after 2010 in both men and women, possibly indicating emerging or under-recognized health challenges within this group. Regarding autocorrelation for Chapter 6, residuals for men showed no significant autocorrelation (*Q** = 14.837, *df* = 10, *p* = 0.138), whereas for women, significant autocorrelation was present (*Q** = 29.943, *df* = 10, *p* = 0.0009). For Chapter 5, residuals for men exhibited significant autocorrelation (*Q** = 20.683, *df* = 10, *p* = 0.0234), while those for women did not (*Q** = 17.852, *df* = 10, *p* = 0.0575). Finally, for Chapter 10, no significant autocorrelation was detected for men (*Q** = 4.125, *df* = 10, *p* = 0.942) or women (*Q** = 6.128, *df* = 10, *p* = 0.804) (Appendix A).

Notably, women exhibited higher mortality rates ($\bar{MR}=87.2$) than men ($\bar{MR}=81.2$) for Chapter 9. For the remaining chapters, men generally showed higher rates. A remarkable temporal shift was observed in Chapter 4: while women initially had higher mortality rates (${\bar{MR}}_{2000-2008}=52.9$), male mortality (${\bar{MR}}_{2009-2019}=71.88$) surpassed that of females after 2008 (Figure 3).

#### 3.1.3. Age-Specific Mortality Trends

By analyzing age group-specific mortality trends, significant negative trends were found for older adults (65+) for Chapter 4 and Chapter 10 diseases, as well as for children aged 0–4 for Chapter 10. In contrast, Chapter 9 showed positive and statistically significant trends for the age groups 55–64 and 65+. No significant trends were observed for the remaining age groups across any chapter (Table A1). For all disease chapters, age groups above 35 years consistently exhibited the highest mortality rates, with individuals aged 65 and older being the most affected. This pattern likely reflects the cumulative effect of chronic health conditions and increased physiological vulnerability in older adults. Notably, for Chapter 10, children aged 0–4 showed higher mortality rates than individuals aged 5 to 54, suggesting a specific vulnerability at early ages, potentially linked to congenital conditions, inadequate access to early medical care, or socioeconomic disparities (Figure 4).

To assess temporal autocorrelation in the residuals of the age-specific linear regression models, Ljung–Box tests were conducted for all 41 unique combinations of age group and disease chapter. Of these, 13 models (31.7%) exhibited statistically significant autocorrelation in their residuals (*p* < 0.05) (Appendix A).

These findings revealed that autocorrelation was particularly prevalent in certain disease chapters and age groups: Chapter 4 showed significant autocorrelation across multiple adult age groups: 25–34 years (*Q** = 37.7, *df* = 10, *p* < 0.005), 35–44 years (*Q** = 24.6, *df* = 10, *p* = 0.01), 55–64 years (*Q** = 32.0, *df* = 10, *p* < 0.005), and 65+ years (*Q** = 38.8, *df* = 10, *p* < 0.005). For Chapter 9, significant autocorrelation was found in the youngest age group (0–4 years: *Q** = 26.3, *df* = 10, *p* < 0.005) and among adolescents/young adults (15–24 years: *Q** = 24.7, *df* = 10, *p* = 0.01). For Chapter 10, significant autocorrelation was observed in the younger age groups, 0–4 years (*Q**= 22.1, *df* = 10, *p* = 0.01), 05–14 years (*Q** = 21.1, *df* = 10, *p* = 0.02), and 15–24 years (*Q** = 19.5, *df* = 10, *p* = 0.03). Chapter 5 showed significant autocorrelation for age groups 25–34 years (*Q** = 18.6, *df* = 10, *p* = 0.05), 35–44 years (*Q** = 21.2, *df* = 10, *p* = 0.02), and 55–64 years (*Q** = 26.5, *df* = 10, *p* < 0.005). Finally, for Chapter 6, one instance of significant autocorrelation was also noted in the oldest age group (65+ years: *Q** = 23.4, *df* = 10, *p* = 0.01).

This pattern of residual autocorrelation highlights that, while linear models provide insight into overall age-specific trends, more complex temporal structures might be present in specific age-disease dynamics, where autocorrelation was significant. A detailed summary of all Ljung–Box test results for each age group and disease chapter model is provided in Appendix A.

### 3.2. Municipal-Level Trends in Non-Communicable Disease Mortality in the MAVM

#### 3.2.1. Mortality Trends for the Total Population

Mapping mortality trends at the municipal level revealed spatial heterogeneity in the evolution of disease-specific mortality between 2000 and 2019. For the total population, no significant trends were detected for Chapters 5 and 6 in any municipality. In contrast, Chapters 4 and 9 exhibited the highest numbers of municipalities with significant increasing trends, with 52 and 45 municipalities, respectively, accounting for more than 50% of the municipalities in the MAVM. Notably, few municipalities exhibited decreasing trends for these disease chapters. Specifically, negative trends were observed in the Benito Juárez borough (Mexico City) and Nextlalpan (State of Mexico) for Chapter 4, and in Acolman (State of Mexico) for Chapter 9.

For Chapter 10, 11 municipalities showed significant positive trends, with 9 located in Mexico City and 2 in the State of Mexico. These municipalities are concentrated in a region known for persistent poor air quality. Notably, the Cuauhtémoc borough of Mexico City, despite being located in the same area of poor air conditions, showed no significant trend. Additionally, 21 municipalities in the State of Mexico presented negative trends for Chapter 10, predominantly located at the periphery of the MAVM, representing the highest number of municipalities with decreasing mortality trends for any chapter (Figure 5). These results provide evidence of spatial heterogeneity in mortality trends within the MAVM over the last two decades.

Regarding the assessment of temporal autocorrelation in the residuals of these municipal-level linear regression models, Ljung–Box tests were performed for all 369 unique municipality–disease chapter combinations. Of these, 46 models (12.5%) exhibited statistically significant autocorrelation (*p* < 0.05). Indicating that, for a subset of municipalities and disease chapters, the linear trend models did not fully capture the underlying temporal dynamics. Conversely, for the vast majority of models (323 out of 369, or 87.5%), the Ljung–Box test indicated no significant autocorrelation, suggesting that the linear trend models provided an adequate fit without evidence of unmodeled temporal structure. A complete summary of all Ljung–Box test results for each municipality–disease chapter combination is provided in Appendix A.

#### 3.2.2. Sex-Specific Mortality Trends

Temporal mortality trends, analyzed by sex and municipality, demonstrated significant heterogeneity across disease chapters. Notably, Chapter 4 showed widespread increasing mortality in 35 municipalities (12 in Mexico City, 23 in the State of Mexico) for both sexes, with an additional 19 municipalities exhibiting sex-specific increases (men: 12, women: 7). Decreasing trends in this chapter were infrequent and mainly observed in women (5 municipalities). Chapter 9 also displayed prevalent increasing trends in 40 municipalities (14 in Mexico City, 26 in the State of Mexico) for both sexes, alongside sex-specific increases suggesting potential male vulnerability in 8 municipalities and female vulnerability in 2 (Figure 6).

In contrast, Chapter 10 was dominated by decreasing mortality trends across 23 municipalities (both sexes: 8, men only: 9, women only: 6). Fifteen municipalities showed increasing trends in Chapter 10 (both sexes: 10, men: 2, women: 3). Mortality in Chapters 5 and 6 remained with no trends in all municipalities. In summary, increasing trends were more characteristic of endocrine and circulatory diseases, while respiratory diseases showed more decreasing trends. Mental and nervous system disorders exhibited no significant temporal changes (Figure 6). The presence of sex-specific trends in several municipalities underscores the importance of considering gender in understanding mortality vulnerabilities.

By assessing temporal autocorrelation using the Ljung–Box test, we evaluated the residuals of 704 sex-specific, municipal-level linear regression models, each corresponding to a unique combination of municipality, disease chapter, and sex. A total of 61 models (8.7%) exhibited statistically significant autocorrelation (*p* < 0.05), suggesting that, for this small subset, the linear models did not fully account for the underlying temporal structure. Importantly, in the vast majority of cases—643 out of 704 models (91.3%)—no significant autocorrelation was detected. This suggests that the linear trend models provided an adequate representation of temporal patterns across most municipality–disease–sex combinations, with no evidence of unmodeled temporal structure. A complete summary of the Ljung–Box test results for each stratum is provided in Appendix A.

#### 3.2.3. Age-Specific Mortality Trends

Temporal trends in mortality at the municipal level, stratified by age group and disease chapter, revealed heterogeneous patterns. By counting the number of municipalities with positive or negative trends, we found that for Chapter 4, young children (0–4 years) predominantly exhibited negative trends. However, a substantial number of municipalities among age groups above 25 years displayed more positive than negative trends. For Chapter 5, a predominantly negative trend in mortality was observed across all age groups. Similarly, for Chapter 6, mortality trends were primarily negative across most age groups (Figure 7).

In Chapter 9, we observed a higher proportion of municipalities with positive mortality trends, mainly among individuals aged 65 years and older. In contrast, Chapter 10 exhibited predominantly negative mortality trends across nearly all age groups, suggesting a general decrease in respiratory disease mortality across most municipalities. It is noteworthy that the 0–4 age group had the highest number of municipalities exhibiting negative mortality trends (Figure 7).

In summary, mortality trends varied across disease categories and age groups. Endocrine, nutritional, and metabolic diseases (Chapter 4) exhibited a mixed pattern, with younger groups generally showing positive trends, but declines emerging among adults aged 55 years and older. Mental and behavioral disorders (Chapter 5), diseases of the nervous system (Chapter 6), and respiratory-system diseases (Chapter 10) predominantly showed declining mortality trends across all age groups. In contrast, diseases of the circulatory system (Chapter 9) exhibited a concerning pattern of increasing mortality, particularly among older adults (Figure 7).

Temporal autocorrelation in the residuals of the 1876 age-specific, municipal-level linear regression models was assessed using Ljung–Box tests. The vast majority of these models (1698 of 1876; 90.5%) showed no statistically significant autocorrelation (*p* ≥ 0.05), indicating that the linear trend adequately captured the temporal patterns in these cases. However, 178 models (9.5%) exhibited significant autocorrelation (*p* < 0.05), suggesting the presence of unexplained temporal structure that was not accounted for by the linear model. A full summary of the Ljung–Box test results for each combination is provided in Appendix A.

### 3.3. Relative Risk Trends in Non-Communicable Disease Mortality in the MAVM

#### 3.3.1. Sex-Specific Relative Risk of Mortality over Time

The analysis of relative risk (RR) by sex consistently indicated that men had a higher risk (RR > 1, LCL > 1) of mortality across most disease chapters. Chapter 9 was an exception, where women exhibited a higher mortality risk (RR > 1, LCL > 1). However, over the study period, we observed a general decreasing trend in RR among women, whereas for men, the RR showed an increasing tendency. Furthermore, in Chapter 4, a shift in mortality risk between sexes was noted from 2007, where men surpassed women in mortality risk, coupled with a sustained positive trend in mortality due to these conditions (Figure 8)

#### 3.3.2. Age-Specific Relative Risk (RR) of Mortality over Time

The temporal trends of the log-transformed relative risk of mortality (relative to the 25–34 age group) by disease chapters are presented in Figure 9, stratified by age group (0–4, 5–14, 15–24, 35–44, 45–54, 55–64, and 65+ years). In the 0–4 age group, higher RR values (>1) were observed for Chapters 4, 6, and 10, although with a decreasing trend over time. For the 5–14 and 15–24 age groups, we found no significant RR for any disease chapter (RR < 1 and CI < 1), indicating that their mortality rates are lower than the 25–34 reference group. Conversely, older groups (>34 years) exhibit an RR > 1 for most disease chapters, indicating that they have higher mortality rates than the reference group. The 35–44, 45–54, and 55–64 age groups showed high RR values (>1) for Chapters 4 and 9. In the 65+ age group, we found high RR values for Chapters 4, 9, and 10. Across all age groups, Chapter 6 consistently exhibited the lowest RR values (<1).

The analysis reveals significant heterogeneity in the relative risk of mortality across age groups and disease chapters over time. Younger age groups (especially 0–14) generally exhibit lower relative risks compared to the 25–34 reference group for most disease chapters. As age increases, the relative risk of mortality generally rises across all disease categories, with diseases of the circulatory system (Chapter 9) consistently showing the highest relative risk in older adults. The temporal trends vary by age group and disease chapter, suggesting complex interactions between age, cause of death, and time.

#### 3.3.3. Municipality-Specific Relative Risk (RR) of Mortality over Time

To identify spatial patterns of elevated mortality risk, we computed the number of years in which each municipality showed a statistically significant relative risk (RR > 1 and LCI > 1) for each disease chapter between 2000 and 2019. The resulting frequency maps reveal municipalities that exhibited high frequencies of significant RR > 1 (based on 95% confidence intervals, where LCI > 1), indicating persistent excess mortality risk over the study period. These hotspots were particularly concentrated in Mexico City for Chapters 4, 6, 9, and 10. Figure 10 labels the municipalities that had between 15 and 20 years with a significant RR. In contrast, for Chapter 6, municipalities showed few (<10) or no years with significant RR (i.e., where the 95% CI did not exclude 1)**,** suggesting relatively lower-than-expected mortality risk.

The spatial cluster for Chapter 4 included five boroughs of Mexico City: Gustavo A. Madero, Cuauhtémoc, Iztapalapa, Iztacalco, and Azcapotzalco. Chapter 6 comprised three boroughs: Gustavo A. Madero, Miguel Hidalgo, and Benito Juárez. Chapter 9 encompassed nine boroughs: Gustavo A. Madero, Azcapotzalco, Miguel Hidalgo, Cuauhtémoc, Venustiano Carranza, Benito Juárez, Iztacalco, Álvaro Obregón, and Coyoacán. For Chapter 10, the cluster included five boroughs (Gustavo A. Madero, Cuauhtémoc, Miguel Hidalgo, Benito Juárez, and Álvaro Obregón) and two municipalities from the State of Mexico (Naucalpan de Juárez and Villa del Carbón), the latter being the only non-Mexico City municipalities identified as mortality hotspots. This spatial and temporal aggregation summarizes the municipalities with recurrent mortality inequalities and highlights priority areas for further investigation and potential intervention. For example, the cluster for Chapter 9 was the largest, and Gustavo A. Madero appeared in all four clusters.

## 4. Discussion

Our results provide evidence of spatiotemporal patterns in NCD mortality trends within the MAVM over the past two decades, examined across two spatial scales and stratified by total population sex and age groups. At the MAVM level, a predominant upward trend in mortality rates was observed across most NCD chapters during our study period, consistent with global trends reported in low- and middle-income countries undergoing demographic and epidemiological transitions and suggestive of a growing burden of non-communicable diseases [26,27].

A strong positive trend was observed mainly for Chapter 4 (Endocrine, Nutritional, and Metabolic Diseases) and Chapter 9 (Diseases of the Circulatory System). These upward trends may be linked to population lifestyles, obesity, and hypertension as important contributing factors to this increase [27,28]. This pattern is consistent with increasing rates of cardiovascular mortality reported in urban areas of Latin America [28], where Mexico has stood out for the increase in mortality from these conditions [29].

For Chapter 6 (Diseases of the Nervous System) and Chapter 10 (Diseases of the Respiratory System)**,** a moderate upward trend was observed. For Chapter 6, this trend was accompanied by significant autocorrelation in the model residuals, suggesting additional unmodeled temporal patterns. While these trends were influenced by localized or group-specific decreases (as previously shown in Section 3.1.3 and Section 3.2.1)**,** the absolute number of people affected by these conditions increased. For Chapter 6, there is an increase of 75% from 2000 to 2019, whereas for Chapter 10, the increase was 45%. These increases may reflect growing and aging populations or a possible link to environmental stressors [26]. Specifically, for Chapter 10, which includes chronic respiratory conditions, the observed positive trend in mortality could be influenced by exposure to poor air-quality episodes, such as higher levels of particulate matter (PM) and gaseous pollutants (ozone, nitrogen dioxide, and sulfur dioxide) [30], which remain elevated in many municipalities within the Metropolitan Area of the Valley of Mexico [31].

In contrast, the lack of significant change in Chapter 5 (Mental and Behavioral Disorders) could be reflective of underreporting, diagnostic limitations, or the chronic and often non-lethal nature of many psychiatric conditions. The relatively low *R*^2^ (0.15) suggests high interannual variability and possible data noise; however, a 25% increase in deaths was observed from 2000 to 2019. This may reflect a lack of progress in diagnosis, care access, or data capture for these diseases, which have historically been underprioritized in public health agendas [32,33]. These findings emphasize the importance of tailored public health strategies. Targeted prevention and early detection efforts for metabolic and circulatory diseases appear especially urgent, while further investigation into rising trends in neurological and respiratory conditions is warranted.

Temporal mortality trends by ICD-10 chapter, stratified by sex, revealed strong positive trends for both sexes for Chapters 4 and 9, consistent with the overall population-level patterns. For Chapter 9, however, significant temporal autocorrelation was detected in these sex-specific models, indicating the presence of underlying temporal dynamics not fully captured by the linear trend, which deserve future attention. Notably, women exhibited consistently higher mortality rates than men for Chapter 9, with these differences primarily attributable to deaths in older women (65+), similar to recent observations of cardiovascular disease patterns among older women in other countries [34,35,36,37]. However, in recent years (since 2017), this disparity has been narrowing, as reflected by a declining trend in RR values for women and increasing RR values for men. By 2018 and 2019, RR values have closed to 1, suggesting convergence in mortality risk between sexes. In contrast, a marked shift was detected for Chapter 4, where men’s mortality rates surpassed those of women after 2009. This change may reflect a complex interplay of environmental and behavioral factors, higher exposure to metabolic risk factors among men, lower healthcare engagement, or sex-specific differences in access to early diagnosis and treatment [38]. These hypotheses warrant further investigation in future research.

In contrast, no clear temporal trends were identified for Chapters 5 (Mental and Behavioral Disorders), 6 (Diseases of the Nervous System), and 10 (Diseases of the Respiratory System) in either sex. Nonetheless, a slight upward trend was observed in Chapter 6 after 2010 for both men and women, which could indicate increasing exposure to neurotoxic environmental pollutants [39,40] or extreme environmental temperatures [41,42]. These findings underscore the dynamic nature of mortality trends and the importance of incorporating sex-disaggregated and disease-specific analyses into public health surveillance. Moreover, the influence of social determinants—such as access to healthcare, socioeconomic status, and environmental exposures—should be considered when interpreting these patterns and designing targeted interventions [43,44].

Stratifying mortality by age groups revealed that individuals over 25 years exhibited the highest mortality rates, with individuals aged 65 and older experiencing the greatest burden. This pattern is consistent with the accumulation of chronic health conditions and increased vulnerability among older adults, underscoring the need for sustained preventive and clinical care. A distinct pattern emerged in Chapter 10, where children aged 0–4 years had higher mortality rates than those aged between 5 and 54 years. This early-life vulnerability may reflect congenital disorders, nutritional deficiencies, limited healthcare access, or the compounded effects of poverty and environmental exposures. These findings emphasize the importance of addressing social determinants of health across the life course, particularly the need for early childhood interventions in underserved communities to improve early-life health outcomes, especially with regard to respiratory disease [45].

Visual inspection of age-specific mortality trends by ICD-10 chapter (Figure 4) revealed substantial year-to-year fluctuations rather than consistent temporal trends across most age groups. Notable exceptions included increasing mortality trends for the 55–64 and 65+ age groups in Chapter 9 and decreasing trends for the 0–4 and 65+ groups in Chapter 10. In Chapter 4, the 0–4 age group exhibited a pronounced mortality peak in 2002–2003, followed by a decline and another rise from 2007 to 2010, with a sharp decrease in subsequent years. The 5–14 age group also displayed fluctuating patterns, with notable peaks in 2010 and 2014, whereas other age groups showed more stable mortality rates.

In Chapter 5, we observed several years with zero recorded deaths in the 0–4 and 5–14 age groups, suggesting potential underreporting or data sparsity in young children. A bimodal pattern was evident in the 15–24, 25–34, 35–44, and 45–54 age groups, characterized by elevated mortality in the early 2000s (particularly 2000–2004), followed by a decline and a subsequent increase toward 2019. In contrast, the 65+ group exhibited high mortality in the mid-period (2006–2010), peaking in 2014. Chapter 6 showed no clear trends, but rather irregular year-to-year variation across all age groups. In Chapter 9, mortality rates were elevated in the 0–4 and 5–14 groups between 2008 and 2013. For individuals aged 15–24 to 35–44, mortality rates increased steadily from 2011 to 2019, while the 45–54 group maintained relatively stable mortality throughout the study period. A sharp spike in mortality was evident in 2009 among individuals aged 5–44 for Chapter 10, likely reflecting the impact of the H1N1 influenza pandemic on respiratory health. Although similar increases were noted in the 45–54 and 55–64 groups, even higher mortality levels were recorded in 2018 and 2019.

These fluctuations highlight the complexity of age-specific mortality patterns and the challenges in detecting structural trends without more granular or dynamic modeling approaches. However, the findings reinforce the need to strengthen public health policies aligned with Mexico’s Programa Sectorial de Salud 2020–2024 [46], which prioritizes reducing mortality among vulnerable populations—particularly children and older adults—by improving access to quality healthcare services, promoting early diagnosis, and implementing targeted public health interventions [43].

The spatiotemporal dynamics of statistically significant relative risk highlight persistent inequalities in mortality across municipalities. The aggregation of municipalities consistently exhibiting an RR > 1 throughout the entire period likely reveals those that experienced sustained structural or environmental conditions contributing to elevated mortality risks from 2000 to 2019. By highlighting municipalities with 15 or more years of high RR values, we identified four spatiotemporal mortality hotspots for ICD-10 chapters 4, 6, 9, and 10, primarily located in Mexico City. The identification of these persistent mortality hotspots in Mexico City (Figure 10) is robust, as these areas consistently demonstrated statistically significant elevated relative risks (95% LCI > 1) over multiple years, underscoring areas where interventions are most urgently needed. The boroughs that stand out in these hotspots are Gustavo A. Madero, which appeared in all four clusters, and Cuauhtémoc and Miguel Hidalgo, each included in three. These boroughs also exhibited increasing trends in mortality rates, underscoring the urgent need for targeted public health interventions in these areas. The cluster associated with Chapter 9 is particularly concerning, as it was the largest among all, highlighting widespread and persistent inequalities related to circulatory system diseases.

Geographical clusters of elevated mortality risk, particularly concentrated within Mexico City boroughs, strongly suggest the influence of localized environmental and social determinants, which often exhibit shared pathways. For instance, the increasing mortality trends for Chapter 10 in several Mexico City boroughs (Figure 5) align with the known challenges of chronic air pollution in these densely populated urban centers [47,48]. The semi-enclosed basin topography and frequent thermal inversions in the MAVM trap pollutants, exacerbating respiratory health outcomes [49].

Furthermore, the significant and persistent mortality risks observed for cardiovascular (Chapter 9) and metabolic (Chapter 4) diseases in these urban hotspots often coincide with areas facing higher socioeconomic disparities [9]. These disparities can manifest through limited access to nutritious food, inadequate opportunities for physical activity, chronic stress, barriers to quality healthcare services, inadequate green spaces, poor housing conditions, and lifestyles influenced by urban sprawl, all of which are well-documented drivers of NCD burden [50]. While our study did not directly quantify the contribution of each determinant, the clustering of elevated risk in these specific boroughs points to the interplay between unhealthy urban environments and socioeconomic vulnerabilities. Future research should prioritize disentangling the individual and synergistic effects of these complex determinants on NCD mortality at fine spatial scales.

These findings underscore the need for geographically-targeted public health interventions and long-term policies that address the structural determinants of health. Furthermore, the clustering of high-risk municipalities across time and disease types indicates that regional strategies may be more effective than isolated municipal efforts [51]. Understanding the spatial persistence of mortality risk not only informs risk prevention and resource allocation but also reflects broader social inequities, thereby reinforcing the importance of equity-focused health policies.

While our analysis offers a detailed spatial and temporal characterization of mortality trends in the Mexico City Metropolitan Area, it is important to acknowledge a few limitations. Firstly, the accuracy of mortality data relies heavily on death certificates; thus, potential misclassification of causes of death could introduce bias, especially concerning chronic diseases. Secondly, analyzing data at the municipal level might mask existing inequalities within those municipalities. Despite these limitations, our findings provide crucial insights into the spatial disparities in mortality dynamics, offering significant implications for public health planning. Identifying municipalities with increasing mortality trends pinpoints areas where disease prevention and control efforts are urgently needed. Furthermore, the persistence of high mortality rates in central urban areas highlights the complex interplay among environmental exposures, social vulnerability, and health outcomes. This understanding is vital for developing targeted and effective public health strategies.

## 5. Conclusions

Our study provides compelling evidence of significant spatial heterogeneity in mortality trends within the Metropolitan Area of the Valley of Mexico over the past two decades. We observed substantial variation in mortality trajectories across ICD-10 chapters, as well as among sex and age groups within each chapter. A predominant upward trend in mortality rates was evident in most NCD chapters at the MAVM level. Between 2000 and 2019, we documented marked increases in deaths for Chapter 9 (Diseases of the Circulatory System) by 97%, Chapter 4 (Endocrine, Nutritional, and Metabolic Diseases) by 97%, Chapter 6 (Diseases of the Nervous System) by 75%, Chapter 10 (Diseases of the Respiratory System) by 45%, and Chapter 5 (Mental and Behavioral Disorders) by 28%. These findings underscore the growing burden of non-communicable diseases in the region and highlight the need for targeted, place-based interventions that address the diverse demographic and epidemiological profiles of the MAVM.

Diseases of the circulatory system (Chapter 9) and endocrine, nutritional, and metabolic diseases (Chapter 4) emerged as particularly concerning, with over 50% of municipalities (45 and 52, respectively) exhibiting increasing mortality trends. The burden of Chapter 9 was especially pronounced among older adults, while for Chapter 4, men’s mortality rates surpassed those of women after 2009. These patterns are critical for identifying specific geographic areas and population subgroups where mortality is either increasing or declining. Such a granular understanding is essential to inform targeted public health interventions and shape future research agendas.

From a policy perspective, our findings highlight an urgent need for evidence-based interventions tailored to the most affected municipalities and vulnerable population groups, specifically: (1) The widespread increase in Chapters 4 and 9 mortality across more than half of MAVM municipalities demands robust, integrated strategies for the prevention and control of metabolic and cardiovascular diseases across the entire metropolitan area. (2) Although Chapter 9 and Chapter 4 exhibited the largest increases in mortality—with the number of deaths rising by 97% between 2000 and 2019—Chapter 6 (+75%) and Chapter 10 (+45%) also experienced substantial growth during this period. The sustained upward trajectories over nearly two decades suggest potential links to environmental stressors (e.g., air pollution, urban heat), aging population structures, and gaps in access to specialized care—all of which require integrative, place-based public health responses. (3) The reversal in sex-specific mortality trends in Chapter 4, where men mortality overtook female mortality post-2009, signals the need for sex-specific public health campaigns and healthcare access initiatives to address male-specific vulnerabilities and behaviors. (4) Geographical clusters highlight areas where interventions are most urgently needed. For instance, Gustavo A. Madero, Cuauhtémoc, and Miguel Hidalgo require specific and urgent attention for metabolic, cardiovascular, respiratory, and neurological diseases. By pinpointing these high-priority areas and specific demographic shifts, public health initiatives can be more effectively designed and implemented to address the unique health challenges of the MAVM.

## Figures and Tables

**Figure 1 diseases-13-00241-f001:**
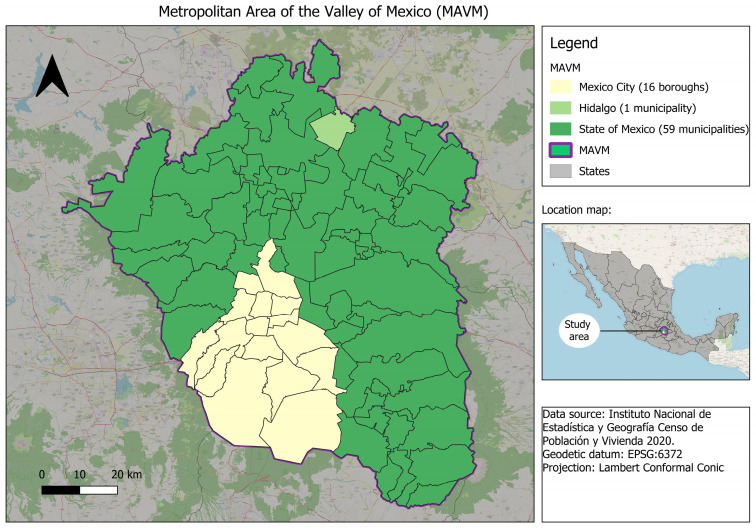
Map of the Metropolitan Area of the Valley of Mexico, showing the boundaries of municipalities included in the study.

**Figure 2 diseases-13-00241-f002:**
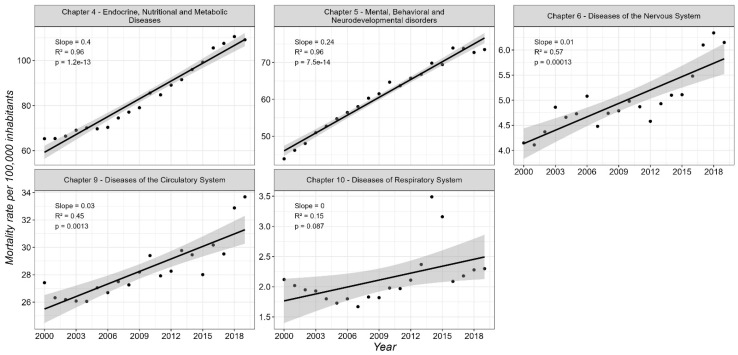
Temporal trends in mortality rates (per 100,000 inhabitants) for non-communicable diseases in the MAVM, 2000–2019.

**Figure 3 diseases-13-00241-f003:**
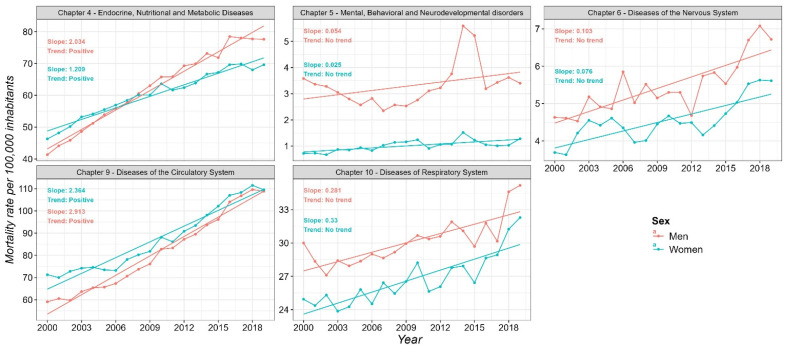
Temporal trends in mortality rates (per 100,000 inhabitants) by sex for non-communicable diseases in the MAVM from 2000 to 2019.

**Figure 4 diseases-13-00241-f004:**
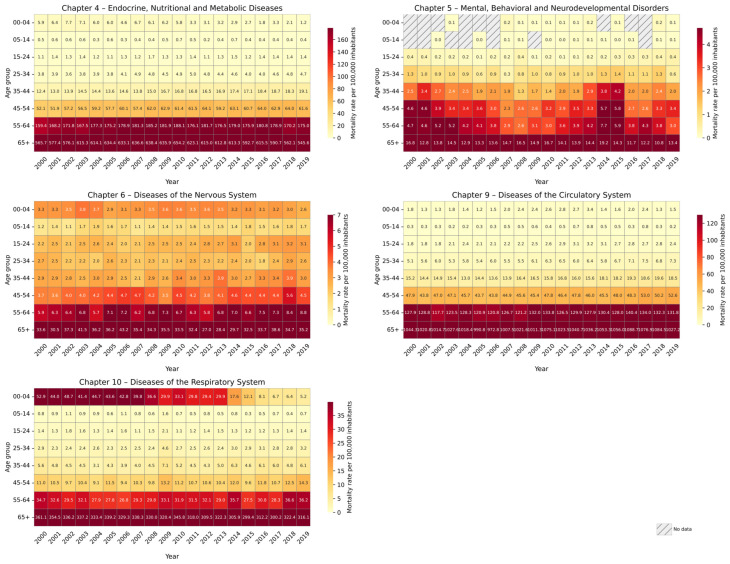
Temporal trends in mortality rates (per 100,000 inhabitants) by age groups for non-communicable diseases in the MAVM from 2000 to 2019.

**Figure 5 diseases-13-00241-f005:**
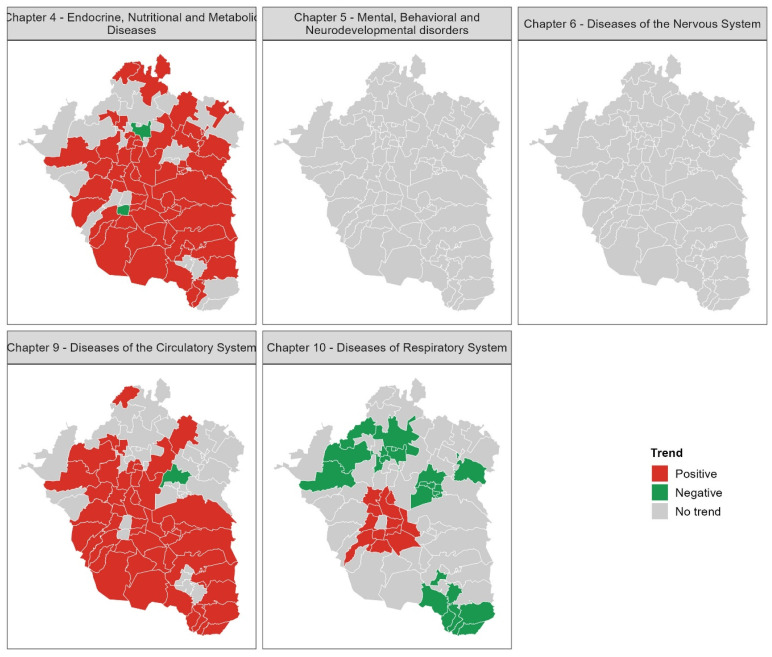
Trends in non-communicable disease mortality across municipalities of MAVM for the total population (2000–2019).

**Figure 6 diseases-13-00241-f006:**
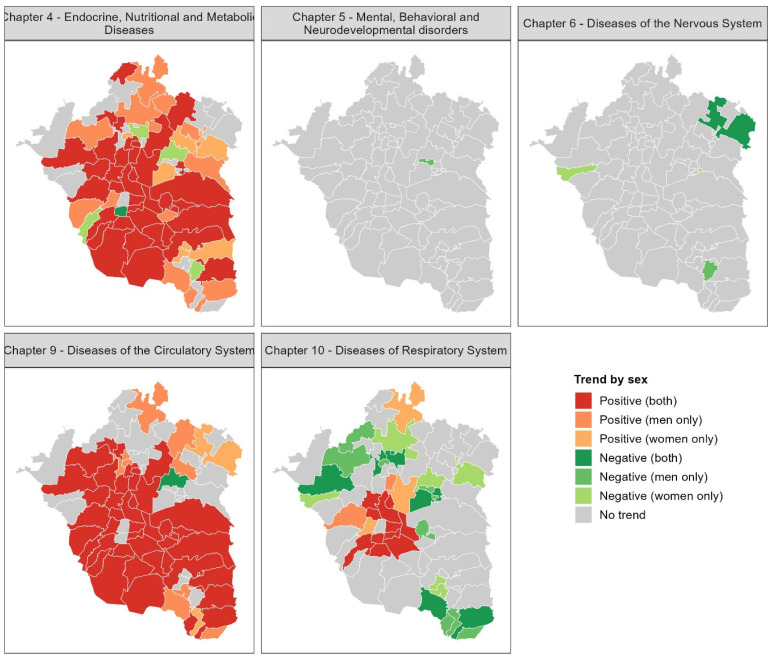
Spatial distribution of mortality trends by sex in the MAVM at the municipal level (2000–2019).

**Figure 7 diseases-13-00241-f007:**
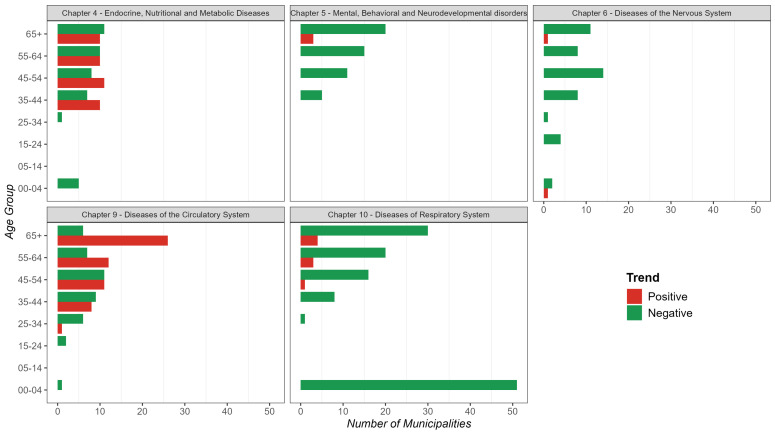
Number of municipalities grouped by temporal trends (positive or negative) in mortality, stratified by age group and ICD-10 disease chapter.

**Figure 8 diseases-13-00241-f008:**
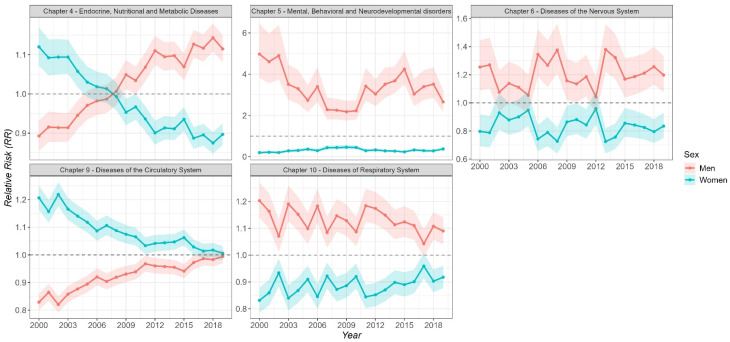
Temporal trends of sex-specific relative mortality risk for non-communicable diseases in the MAVM (2000–2019).

**Figure 9 diseases-13-00241-f009:**
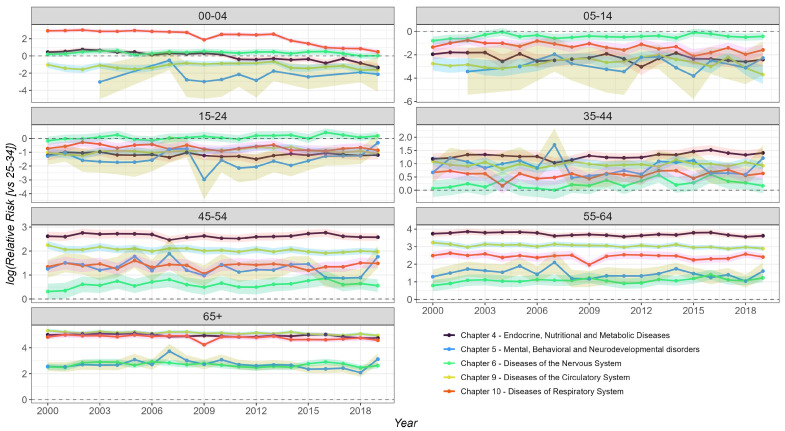
Temporal trends of log-transformed relative mortality risk across ICD-10 disease chapters by age group.

**Figure 10 diseases-13-00241-f010:**
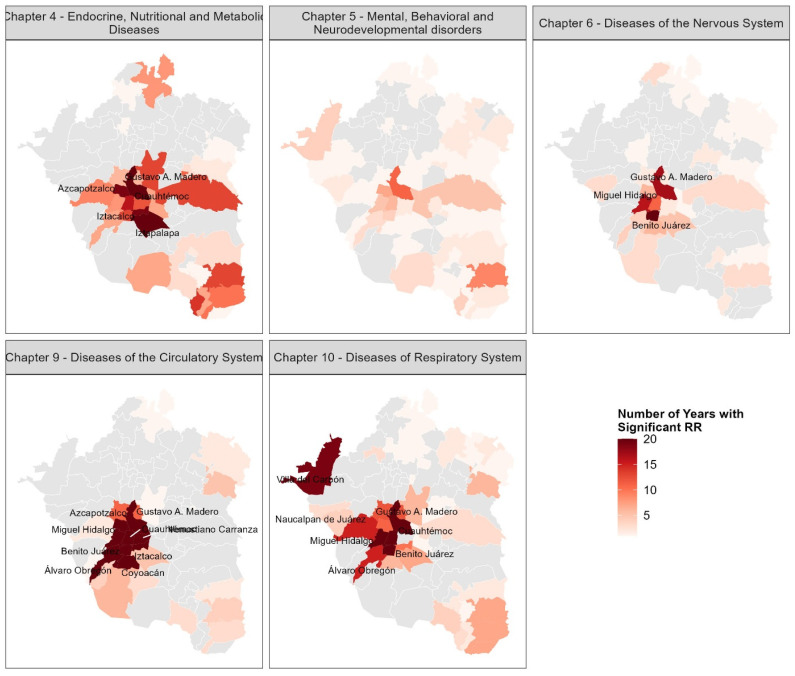
Temporal frequency of statistically significant relative risk (RR > 1 and 95% lower confidence limit > 1) of mortality from non-communicable diseases across municipalities compared to the mortality rate of MAVM, 2000–2019. Labels indicate municipalities with significant RR in ≥15 years.

**Table 1 diseases-13-00241-t001:** Total deaths recorded for five NCD chapters in the Metropolitan Area of Mexico Valley (2000–2019), stratified by sex and age group.

		Cumulative Deaths from 2000 to 2019					
Sex	Age Group	Chapter 4	Chapter 5	Chapter 6	Chapter 9	Chapter 10	Total
Men	0–4	937	12	675	376	6365	8365
	5–14	177	16	624	171	301	1289
	15–24	519	135	1285	1185	638	3762
	25–34	1916	618	972	2923	1344	7773
	35–44	6114	1313	1054	6547	1908	16,936
	45–54	16,500	1457	1060	13,618	2926	35,561
	55–64	28,239	1079	1130	22,913	5222	58,583
	65+	70,865	1977	4099	115,163	41,363	233,467
Women	0–4	743	8	507	309	4709	6276
	5–14	160	10	465	143	254	1032
	15–24	457	35	601	643	407	2143
	25–34	1093	54	645	1341	560	3693
	35–44	3325	72	712	3058	1026	8193
	45–54	10,580	114	869	7561	2013	21,137
	55–64	23,120	134	891	14,509	3879	42,533
	65+	88,320	1733	4906	157,741	43,688	296,388
	**TOTAL**	**253,065**	**8767**	**20,495**	**348,201**	**116,603**	**747,131**

Chapter 4: Endocrine, Nutritional, and Metabolic Diseases; Chapter 5: Mental, Behavioral, and Neurodevelopmental Disorders; Chapter 6: Diseases of the Nervous System; Chapter 9: Diseases of the Circulatory System; Chapter 10: Diseases of the Respiratory System.

## Data Availability

The original data presented in the study are openly available in the General Directorate of Epidemiology repository, [http://www.dgis.salud.gob.mx/contenidos/basesdedatos/da_defunciones_gobmx.html, accessed on 17 November 2024].

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
