# Peer review of "Spatiotemporal Patterns of Non-Communicable Disease Mortality in the Metropolitan Area of the Valley of Mexico, 2000–2019"

_diseases, 2025, doi:10.3390/diseases13080241_

Round 1

Reviewer 1 Report

Comments and Suggestions for Authors

This manuscript presents a comprehensive spatiotemporal analysis of non-communicable disease (NCD) mortality trends in the Metropolitan Area of the Mexico Valley (MAVM) from 2000 to 2019. The authors focus on five major categories of NCDs, classified by ICD-10 Chapters (4, 5, 6, 9, and 10), and analyze mortality patterns by age, sex, and municipality using standardized mortality rates and relative risk (RR) estimations. The study is well-structured, methodologically sound, and makes a significant contribution to the literature on spatial epidemiology and urban health disparities in low- and middle-income countries.

The study addresses a critical gap in public health research in Mexico by integrating spatial and temporal dimensions of mortality data to identify vulnerable populations and mortality hotspots. Given the high burden of NCDs and growing urban populations in Latin America, this work is timely and relevant. The results have strong implications for health policy, particularly in the context of the Sustainable Development Goal (SDG) target 3.4.

In my opinion, the major strengths of the manuscript are:

  1. Robust Methodology: The use of age- and sex-standardized mortality rates, relative risk estimation, and linear regression trend analysis across two spatial scales (metropolitan and municipal) demonstrates methodological rigor.
  2. Granular Analysis: The stratification by sex and age groups, along with the mapping of mortality trends and hotspots, provides detailed insights into local health dynamics.
  3. Policy Relevance: The findings are highly actionable, particularly the identification of municipalities with persistently elevated mortality risk, and the sex- and age-specific trends in disease burden.
  4. Longitudinal Scope: A 20-year analysis offers a meaningful perspective on the evolution of NCD mortality in response to demographic and environmental changes.

On the other hand, please find below my recommendations from areas that have improvement to address them

  1. Clarity in Language: The manuscript would benefit from careful proofreading to correct grammatical and typographical errors, particularly in the abstract and introduction (e.g., "generally are not transmissible" should be "are generally not transmissible").
  2. Discussion of Environmental Determinants: While the authors acknowledge the role of air pollution and social inequalities, these determinants could be discussed more thoroughly, particularly regarding the spatial clustering of mortality.
  3. Presentation of Statistical Results: Although relative risks and trends are well explained, the confidence intervals for municipal-level RR values could be more clearly emphasized in both figures and text.
  4. Conclusion Section: The conclusion would be strengthened by summarizing key quantitative findings (e.g., percentage increase in mortality rates) and more explicitly linking them to actionable recommendations for public health practitioners.
Comments on the Quality of English Language

The manuscript would benefit from careful proofreading to correct grammatical and typographical errors, particularly in the abstract and introduction (e.g., "generally are not transmissible" should be "are generally not transmissible").

Author Response

Comments 1: Clarity in Language: The manuscript would benefit from careful proofreading to correct grammatical and typographical errors, particularly in the abstract and introduction (e.g., "generally are not transmissible" should be "are generally not transmissible").

Response 1: Thank you for pointing this out. We have performed extensive proofreading across all sections of the manuscript (from the abstract to the conclusions) to correct grammatical and typographical errors. All changes are marked in red text. Additionally, our colleague Dr. Christopher Stephens, a native English speaker (British), kindly conducted a thorough language review

Comments 2: Discussion of Environmental Determinants: While the authors acknowledge the role of air pollution and social inequalities, these determinants could be discussed more thoroughly, particularly regarding the spatial clustering of mortality

Response 2: We appreciate this comment. To strengthen this aspect, we have expanded the discussion of environmental and social determinants (Lines 636–663), which we believe has significantly improved the manuscript 

Comments 3: Presentation of Statistical Results: Although relative risks and trends are well explained, the confidence intervals for municipal-level RR values could be more clearly emphasized in both figures and text

Response 2: We thank the reviewer for this suggestion. We have incorporated the confidence interval information into the corresponding results section (3.3.3) and updated the figure captions accordingly, thereby improving the clarity of the results presentation.   

Comments 4: Conclusion Section: The conclusion would be strengthened by summarizing key quantitative findings (e.g., percentage increase in mortality rates) and more explicitly linking them to actionable recommendations for public health practitioners

Response 2: Thank you for this valuable suggestion. We have revised the conclusion section to include key quantitative findings that highlight emerging health concerns related to specific disease chapters, such as diseases of the nervous and respiratory systems. We have also added actionable recommendations based on our findings regarding high mortality risks in certain population groups and municipalities.  

4. Response to Comments on the Quality of English Language

Point 1: The manuscript would benefit from careful proofreading to correct grammatical and typographical errors, particularly in the abstract and introduction (e.g., "generally are not transmissible" should be "are generally not transmissible.

Response 1: We have corrected this grammatical error and performed an extensive revision of the entire manuscript to address all grammatical and typographical issues. This revision was supported by Dr. Christopher Stephens, who is a native English speaker.

Reviewer 2 Report

Comments and Suggestions for Authors

Title: Ok; however, it is better to show the time period of the study in the title as well.

Abstract:

  • According to the proposed design of the study, it is better to show the findings numerically in this case the conclusion will be more sensible for the readers.
  • In the method part of this section, it is essential to hallmark the comparators used for showing the at-risk populations.

Keywords: OK.

Introduction:

  • I think it is essential to show the setting of the study in more detail (such as the geographical area, population and if possible social, economic and political characteristics). I think what is expressed in the following section of method, is not enough with this regard.

Material and method:

  • Please show how the model used for assessing the temporal trends (section 2.4.1) was validated.
  • I think one important issue to be noticed in the analysis of this study, is the matter of auto-correlation of data of each year; which if it is high, it will violate the used regression model. In this case, it is better to show this pattern by using a correlogram.
  • If possible, please show the completeness of the death registration data used for the study.

Results:

  • For a better informative table, please show each disease group, by their name rather than chapter. This is true for figure 2-10.
  • I think the method of figure 8 and 9 are not addressed in the material and method section. This will make the interpretation difficult for the readers.

Discussion:

  • I think there are a number of aberrancies shown in figure 4. For instance, there are sudden increases of the mortalities shown for a number of disease groups in a certain age groups. I these issues shall be addressed in this section.

References: if possible, use more recent references.  

Comments on the Quality of English Language

I think the first word of in the discussion section is typed wrongly. 

Author Response

Comments 1: Title: Ok; however, it is better to show the time period of the study in the title as well.

Response 1: We agree, and the time period has been added to the title

Comments 2: Abstract: According to the proposed design of the study, it is better to show the findings numerically in this case the conclusion will be more sensible for the readers.

Response 2: We agree. Key numerical findings have now been added to the abstract to highlight emerging health issues.

Comments 3: Abstract: In the method part of this section, it is essential to hallmark the comparators used for showing the at-risk populations.

Response 3: We have integrated the reference population groups (women, 25–35 age group) used to identify at-risk populations. In addition, we specified that metropolitan mortality rates served as the reference for identifying municipalities with excess mortality (Lines 21–22).

Comments 4: Introduction: I think it is essential to show the setting of the study in more detail (such as the geographical area, population and if possible social, economic and political characteristics). I think what is expressed in the following section of method, is not enough with this regard

Response 4: We thank the reviewer for this suggestion. We have added a detailed description of the Metropolitan Area of the Valley of Mexico (MAVM) in the Introduction section, emphasizing its national relevance as well as potential environmental and social issues that may influence health outcomes (Lines 83–103).

Comments 5: Material and method: Please show how the model used for assessing the temporal trends (section 2.4.1) was validated

Response 5: Thank you for this comment. As suggested, we used the Ljung-Box test and Autocorrelation (ACF) and Partial Autocorrelation (PACF) plots (correlograms) for the visual and statistical evaluation of the linear models. This analysis has been added to the Methods section (Lines 169–183)

Comments 6: Material and method: I think one important issue to be noticed in the analysis of this study, is the matter of auto-correlation of data of each year; which if it is high, it will violate the used regression model. In this case, it is better to show this pattern by using a correlogram.

Response 6: Thank you for this valuable suggestion. We evaluated autocorrelation in the residuals of all linear models. We found statistically significant autocorrelation in a subset of models. For instance, at the metropolitan level (total population), 2 of the 5 disease chapters (Chapters 9 and 10) showed autocorrelation; for sex-stratified models, 4 of 10; and for age groups, 13 of 41 models showed autocorrelation. At the municipal level, 46 of 369 total population models, 21 of 704 sex-specific models, and 178 of 1876 age-group-specific models showed autocorrelation. Although these findings suggest residual temporal structures not fully captured by the linear models, we retained the linear approach for two main reasons:

(1) Our primary objective was to identify long-term linear trends and quantify their overall magnitude (slope), which these models achieve effectively; and (2) Autocorrelation mainly affects the precision of statistical inference (e.g., standard errors and p-values) rather than the detection of the underlying trend, particularly when R² values are high (e.g., R² = 0.96 for Chapter 9). Nonetheless, we acknowledge that the presence of autocorrelation suggests these results should be interpreted with caution, and future research could benefit from more advanced time-series models to better account for complex temporal dependencies.

Comments 7: Material and method: If possible, please show the completeness of the death registration data used for the study.

Response 7: Thank you for this suggestion. We have added information about mortality data in the supplementary material.

Comments 8: Results: For a better informative table, please show each disease group, by their name rather than chapter. This is true for figure 2-10.

Response 8: Thank you for this valuable suggestion. The full names of the disease chapters have been added to all relevant figures (2–10).

Comments 9: Results: I think the method of figure 8 and 9 are not addressed in the material and method section. This will make the interpretation difficult for the readers.

Response 9: Thank you for your comment. The methods for Figures 8 and 9 are described in Sections 2.4.2 and 2.4.3 (Lines 184–219), which correspond to the calculation of relative risk.

Comments 10: Discussion: I think there are a number of aberrancies shown in figure 4. For instance, there are sudden increases of the mortalities shown for a number of disease groups in a certain age group. I these issues shall be addressed in this section.

Response 10: Thank you for this suggestion. We have added a discussion of the aberrancies present in temporal trends by age groups at the MAVM level (Lines 600–628).

Comments 11: References: if possible, use more recent references.

Response 11: We appreciate this observation. Upon review, we found that 35% of the 52 references are from 2021–2025, and 56% are from 2011–2020. Therefore, 90% of our references are from the last 15 years, which we believe ensures the manuscript is well supported by recent literature.

4. Response to Comments on the Quality of English Language

Point 1: I think the first word of in the discussion section is typed wrongly.

Response 1: We have corrected this grammatical error
